# A critical analysis of the What3Words geocoding algorithm

**Rudy Arthur** ⓘ *

Department of Computer Science, University of Exeter, Exeter, United Kingdom

* r.arthur@exeter.ac.uk

## Abstract

What3Words is a geocoding application that uses triples of words instead of alphanumeric coordinates to identify locations. What3Words has grown rapidly in popularity over the past few years and is used in logistical applications worldwide, including by emergency services. What3Words has also attracted criticism for being less reliable than claimed, in particular that the chance of confusing one address with another is high. This paper investigates these claims and shows that the What3Words algorithm for assigning addresses to grid boxes creates many pairs of confusable addresses, some of which are quite close together. The implications of this for the use of What3Words in critical or emergency situations is discussed.

**Data Availability Statement:** The data and code used in this study are publicly available from the Zenodo repository (https://doi.org/10.5281/zenodo.8257690) and from the Github repository (https://github.com/rudyarthur/w3wanalysis).

## Introduction

Geocoding is the process of assigning a co-ordinate pair, usually latitude and longitude, given a textual description of a location. Reverse geocoding is the opposite, converting co-ordinates to descriptions. Usually these text descriptions are standard street addresses and/or postal codes where the geocoding process can be quite involved and potentially ambiguous [1]. The company What3Words (hereafter W3W) has proposed an alternative addressing system which they claim 'solves' the geocoding problem. Quoting W3W [2]: *We have divided the world into 3m squares and given each square a unique combination of three words. what3words addresses are easy to say and share, and as accurate as GPS coordinates.*

$$51.520847, -0.19552100 \leftrightarrow ///filled.count.soap$$

The mapping above shows a W3W address (the W3W corporate office in central London) and its equivalent GPS coordinates.

W3W report various applications of their system in logistics, taxi services and navigation [3]. W3W also share numerous instances of use of the app by emergency services, notably the claim that W3W is used by 85% of emergency services in the UK [4]. While claims from W3W themselves may be treated with some scepticism, it is the case that major organisations like car breakdown service the AA [5] and the London ambulance service [6] encourage and give advice on using W3W for reporting locations. Other agencies, such as Mountain Rescue England and Wales, advise using it only in conjunction with other location methods [7].

It is for applications in emergency situations that W3W has been the target of robust criticism in the popular literature for, among other issues, its closed-source code [8] and the

**Funding:** The author received no specific funding for this work.

**Competing interests:** The authors have declared that no competing interests exist.

potential for confusion between addresses [9–11]. A website devoted to issues arising from use of W3W has been established at https://w3w.me.ss/. If W3W is widely adopted by emergency services it must be subject to rigorous evaluation. To my knowledge there has been no study of W3W in the academic literature, though there are some papers on extensions of W3W for 3-dimensional addressing [12, 13] and independent researchers have published some quite detailed analysis on blogs [9, 10]. Here I aim to perform a rigorous evaluation of W3W by analysing their geocoding algorithm.

The outline of the paper is as follows: first the W3W geocoding and reverse geocoding algorithms are described. I then present a model of address transmission, with application by emergency response in mind, and discuss some of the errors which might occur when transmitting W3W addresses. This model is used to study first the prevalence of confusing addresses anywhere in the world and then confusing addresses which are close together. Based on the results of this analysis I offer some advice on the use of W3W and some suggestions for future work. W3W is a commercial entity and does not provide the word list used to make addresses and also limits the number of addresses which can be requested for free. Due to these restrictions, rather than sampling many W3W addresses from the real application this paper will rely on the description of the algorithm given in the patent [14] and use a different (though likely highly overlapping) word list.

## The What3Words algorithm

A user of the W3W service simply interacts with the mobile application or website https://what3words.com/ either by entering a W3W address to get its location on a map (geocoding) or tapping the locate button to obtain their W3W address (reverse geocoding). The details of transforming between geodesic coordinates and W3W addresses are hidden from the user and the assignment of words to locations is fixed. It is the process of assigning word triples to locations, as described in the What3Words patent [14], that will concern us here.

The globe is first partitioned into $4320 \times 8640$ lat(itude) and lon(gitude) cells. These cells are given integer co-ordinates $X, Y$. Each cell is further subdivided into 1546 latitude bands and a variable number (between 1 and 1546) of longitude bands, giving small, approximately $3m \times 3m$, boxes with integer co-ordinates $x, y$. In detail

$$
\begin{aligned}
X &= \text{floor}((\text{lon} + 180) \times 24) \\
Y &= \text{floor}((\text{lat} + 90) \times 24) \\
x &= \text{floor}((W(Y) \times \text{frac}((\text{lon} + 180) \times 24)) \\
y &= \text{floor}((1546 \times \text{frac}((\text{lat} + 90) \times 24)) \\
W(Y) &= \max(1, \text{floor}(1546 \times \cos((Y + 0.5)/24 - 90)))
\end{aligned}
\tag{1}
$$

These coordinates are converted into a number

$$
n = q + 1546x + y
\tag{2}
$$

Where $q$ is an offset which is a function of $X, Y$, chosen such that popular locations (e.g. central London) have smaller $n$ values. To avoid a regular assignment of words to cells, the values of $n$ are then used as input for a linear congruence. A word list of $L = 40,000$ unique terms are assigned into one of $N_b$ bands (where [9] suggests $N_b = 16$ in practice). Each $XY$-cell is also assigned to a band and each band is transformed differently. The value $n$ is transformed to $m$ via the congruence

$$
m = c_b + (a_b \times n) \bmod K^3
\tag{3}
$$

where $K = L/N_b$ and $c_b$, $a_b$ are band-specific constants. Band zero, corresponding to the most popular locations, has $c_b = 0$. For simplicity I will mostly consider band 0. To produce a 3 word address, $m$ is factored into a triple of integers $i, j, k$. To do this, first compute $l = $ floor $(m^{1/3})$. Depending on the value of $\max(i, j, k)$ the factorisation algorithm will be different. For illustration, if $i = \max(i, j, k)$ then one sets $i = l$ and uses

$$m = l^3 + (l + 1)j + k$$

to find $j$ and $k$. Full details for different factorisations are given in the patent [14], but it is always the case that $l = \max(i, j, k)$.

The transformation from $m$ to $n$ in the first band is $m = (a_b \times n) \bmod K^3$. Therefore, the maximum value of $m$ is $K^3 - 1$ and hence $\max(i, j, k)$ can be at most $K - 1$. This means only $K$ words, rather than $L$, are used per band. The words in the word list are indexed such that short, common words have smaller indices and are in lower bands, so the low bands are used for major population centers and higher bands for uninhabited places or the ocean. Finally, the triple of integers $i, j, k$ are used as indices for the word list to form a triple of words which is a W3W address.

To go from W3W addresses to lat, lon coordinates the algorithm above is applied in reverse to obtain $X, Y, x, y$ and lat, lon are then given by inverting Eq 1

$$\begin{aligned} \text{lat} &= (Y + (y + 0.5)/1546)/24 - 90 \\ \text{lon} &= (X + (x + 0.5)/W(Y))/24 - 180 \end{aligned}$$

## Transmission model

I use the **noisy channel model** [15, 16] for the transmission of W3W addresses. A location of interest to the sender is given a W3W address, $s = u.v.w$ which is communicated to a receiver over a noisy channel. The noisy channel induces a change in the message to $s' = u'.v'.w'$. It can be that $u' = u$ etc. but if there is at least 1 difference between $s$ and $s'$ then call $s'$ a **confusion** of $s$. The idea here is to model a situation where someone looks up the W3W address for their location (reverse geocoding) and transmits it to a third party who transforms it back to latitude and longitude (geocoding). I assume no errors in the reverse geocoding/geocoding steps (carried out via the algorithm above) but that some errors could be induced in the transmission process. For example, if a lost hiker phones the emergency services the dispatcher must accurately record the address and pass it on to search and rescue operators who then use it to find the target location. The noisy channel is everything that happens in between the hiker getting a W3W address and the search and rescue team inputting it into the geocoder.

## Confusion modes

There are many potential ways a W3W address could be miscommunicated, a non-exhaustive list is given in Table 1. Due to the restrictive terms of W3W user license [18] it is difficult to empirically investigate the frequency of each of these error modes. The word list is not public and only a limited number of addresses are available without purchasing privileged access. For simplicity in this work I will only explicitly consider typographic errors and simple homophones and therefore the results presented here should be thought of as lower bounds or a best case scenario.

Following similar models for spelling correction define the **confusion set** $C(w)$ as the set of words in a dictionary $\mathcal{D}$ which are 'close' to $w$. For example, it has long been noted [19] that the most common typing errors are

**Table 1. A list of potential error types relevant for transmission of W3W addresses.** These are not necessarily mutually exclusive categories.

| Error Type | Example |
| --- | --- |
| Typing errors | 'fog' becomes 'fig' |
| Homophones | 'their' becomes 'there' |
| Incorrect word form | 'break' becomes 'broke' |
| Autocorrect substitution | 'gunna' becomes 'gunman' [17] |
| Regional spelling variation | 'color' (US) versus 'colour' (UK) |
| Regional homophony | 'tree' and 'three' are homophones in my accent but not in other accents |
| Boundary uncertainty | 'dog.start' versus 'dogs.tart' |
| Multi-word homophony | 'pink.start' versus 'pinks.start' |
| Word transpositions | 'dark.small.places' becomes 'small.dark.places' |
| Translation errors | 'limer.achat.écrire' will not refer to the same location as 'file.purchase.write'. |
| Grammatical gender | e.g. in French, confusing 'avocat' with 'avocate' |
| Adjective agreement | 'chaise.bleu' versus the grammatically correct 'chaise.bleue' |
| Dropping accents | 'ferme' verses 'fermé' |
| Forcing correct grammar | The correct address 'only.neatly.grass' versus the gramatically correct 'only.neat.grass' |

1. transposition of two adjacent letters: 'calm' → 'clam'

2. one extra letter: 'cat' → 'cats'

3. one missing letter: 'broom' → 'boom'

4. one wrong letter: 'frog' → 'from'

For verbally communicating W3W addresses, likely the most problematic words are synophones (word pairs that sound almost the same but have different meanings), 'will' and 'well' seem more likely to be confused than 'will' and 'kill'. However W3W addresses must always be typed for geocoding, so all of the above typing errors are relevant.

Homophones are word pairs like 'which' and 'witch' with similar pronunciation but different spelling. Research confirms that writing a homophone of the intended word is common [20] with the more frequent word most often substituted for the intended one [21] e.g. 'raise' is much more likely to be written than 'raze'. The lack of context in W3W addresses means that the potential for homophone confusion is high, as acknowledged by W3W [22]: *When we select the words to be used in each language, we do our best to remove homophones and spelling variations*. However in practice many can be found and, as mentioned in Table 1, homophony is dependent on accent and it is unlikely that all homophones across all regional accents, even in the UK, can be removed.

To specify the confusion model more precisely, given a dictionary $\mathcal{D}$ of English words, take a sample of unique words from the dictionary to generate a word list $\mathcal{L} = \{w_1, w_2, \ldots, w_L\} \subseteq \mathcal{D}$ which contains $L$ unique words. For English W3W, $L = 40,000$. A W3W address is a triple of words $u.v.w$ where $u, v, w \in \mathcal{L}$. Ignoring complex error modes (see Table 1) and assuming that each word is transmitted independently, the confusion set of an address is the product of the confusion sets of its words (with the convention $w \in C(w)$): $C_\Pi(s) = C(u) \times C(v) \times C(w) - s$. For example, with only homophony as an error mode, the address s = *arose.recede.home* has a possible confusion set: {*arrows.recede.home, arose.reseed.home, arrows.reseed.home*}. If only one word is allowed to be confused, then the confusion set is $C_\Sigma(s) = C(u) \times v \times w + u \times C(v) \times w + u \times v \times C(w) - 3s$. For s = *arose.recede.home* this would give $C_\Sigma(s) = $ {*arrows.recede.home, arose.reseed.home*}. Let the size of the confusion set of $s$ be denoted $c_\Pi(s)$ and $c_\Sigma(s)$ when

allowing many, or only one, word changes, respectively. I will compute both to show that qualitative conclusions do not depend on the choice of $C_\Pi$ or $C_\Sigma$.

## Word list

W3W terms of service prohibit using their word list, so the word list from [23] is used instead. Any list of English words will suffice, this one is comprehensive enough, freely available and words are ranked by frequency of occurrence. Only words at least 4 characters long are kept and words are sampled in decreasing order of frequency with only lower case words which appear at least 3 times in the corpus of [23] are kept. Some words appear with variant spellings, e.g. 'favor' and 'favour'. Since these would likely be removed by W3W I also remove 284 of these from the list, keeping the most common variant. Some examples of the least common words included are 'marinade', 'replayed', 'confusions', by no means difficult or obscure words. Choosing a frequency threshold of 3 results in $L = 43,320$ words, which is close to the number used by W3W, 40,000. Because this list and W3W use common English words, it is likely that both lists have substantial overlap.

To deal with homonyms I use the phonetic dictionary from [24] which gives the International Phonetic Alphabet (IPA) transcription of most English words. Stress markers, used to differentiate e.g. 'in-CREASE' (noun) from 'IN-crease' (verb) are ignored as well as the rare diacritics used to specify the pronunciation of loan words e.g. 'croissant'. The UK and US dictionaries were merged, giving a list of possible pronunciations of each word, then each pronunciation of each word was checked against every other to produce a list of possible homonyms for every word in $\mathcal{D}$. These IPA transcriptions only include standard pronunciations so the number of homonyms found here is a lower bound on homonyms in $\mathcal{D}$ given the wide variety of regional English accents and dialects.

There are 1699 words in $\mathcal{D}$ that have at least one homonym also in the dictionary. Many of these are unusual words e.g. 'sitz'. Restricting the set to only count words in $\mathcal{D}$ that are homophonous with common words ($\mathcal{D}(150)$ see next section for the definition) gives 650 words with common homophones e.g. 'sense' and 'cents'. The word list used by W3W is pruned in some unknown way so these numbers are only estimates. However, anecdotal evidence e.g. [10] and experimentation with the app suggests it is not hard to find addresses in W3W containing a word with a homophone, for example 'arrows.midst.sense' is a valid address in Queensland, Australia.

## Building confusion sets

It will be useful to define $\mathcal{D}(v)$ to be a word list where each word occurs $v$ times in the corpus of [23]. $\mathcal{D}(150)$ gives 7,445 words (roughly the same number as in the first three bands) and $\mathcal{D}(3) = \mathcal{L}$ is the entire word list of 43,320 words. $\mathcal{D}(150)$ will be used to enforce the condition that confusions are with common, rather than rare words. To generate the list of potential confusions of a word $w$:

1. Generate the set of all strings that could be formed by making the typing errors listed above to make $C_t(w)$

2. Look up the set of homonyms of $w$ to get $C_h(w)$

3. Include only common words to give the final confusion set
   $C(w) = (C_t(w) \cup C_h(w)) \cap \mathcal{D}(150)$

For example take $w$ = 'clause'. To make $C_t(w)$ generate mutations of $w$ using the typing error modes described in [19]:

1. Transposition, generating e.g. 'cluase'

2. Adding an extra letter e.g. 'clauses'

3. Deleting a letter e.g. 'cause'

4. Swapping a letter e.g. 'claude'

There is only one homonym of 'clause' in the word list so $C_h(w)$ = {claws}. The union of $C_t(w)$ and $C_h(w)$ is then intersected with $\mathcal{D}(150)$ which removes invalid words (like 'cluase') and rare words. The final result is $C(clause)$ = {clause, cause, claws} where 'cause' is a potential mistyping and 'claws' is a homonym. The potential mis-typing 'clauses' is not included because it fails to meet the frequency threshold.

## Confusing addresses globally

W3W state *The overwhelming proportion of similar-sounding 3 word combinations will be so far apart that an error is obvious* [25]. We will study if the W3W algorithm places similar addresses far apart in the next section, but even if true this type of confusion could still be problematic. Correcting an incorrect address relies on the AutoSuggest algorithm implemented in the W3W application [26] which displays 3 suggested addresses based on the user's input. The algorithm used is closed source and the description in [26] only mentions 'different factors' accounting for which addresses are suggested. However they do state: *In certain situations, like an e-commerce checkout, we increase the likelihood of an accurate match by clipping the area of results to the shipping country*. From these statements, personal experimentation and information given by the help page [27], AutoSuggest seems to balance address similarity with prior information about places and seems to be most likely to make suggestions of populated places near the user. In the worst case, our lost hiker scenario, with the sender in a remote location far from receiver, if the intended address has more than 3 possible confusions, it may not be in the three results displayed.

To model this, assume the probability of a word having $c$ confusions be given by a Poisson distribution with mean $\lambda_1$

$$p(c) = \frac{\lambda_1^c e^{-\lambda_1}}{c!}$$

The probability a word has no confusions is $p(0) = e^{-\lambda_1}$ and the probability that $n$ independent words have no confusions is $(p(0))^n = e^{-n\lambda_1}$. Therefore the probability that an n-tuple has at least 1 confusion is

$$p_n(c > 0) = 1 - e^{-n\lambda_1} \qquad (4)$$

Note this increases exponentially to 1 (certainty of at least one confusion) with $n$ and $\lambda_1$. For W3W $n$ = 3 and $\lambda_1$ is the average number of confusions per word, which depends on the types of error considered.

Using the word list and confusion model (typographic and homonym errors) described, Fig 1(a) shows that the distribution of $p(c)$ is reasonably well approximated by a Poisson with $\lambda_1$ = 0.73 (the orange line). Fig 1(b) estimates the distribution of $p_3(c)$ by generating $10^6$ random word triples. The distributions have mean $\lambda_3$ and $p_3(0)$ = 0.33, meaning only around a third of triples have no possible confusions under the error modes considered here. Thus most addresses, roughly two-thirds, could be confused, by mis-typing or homophony, with another

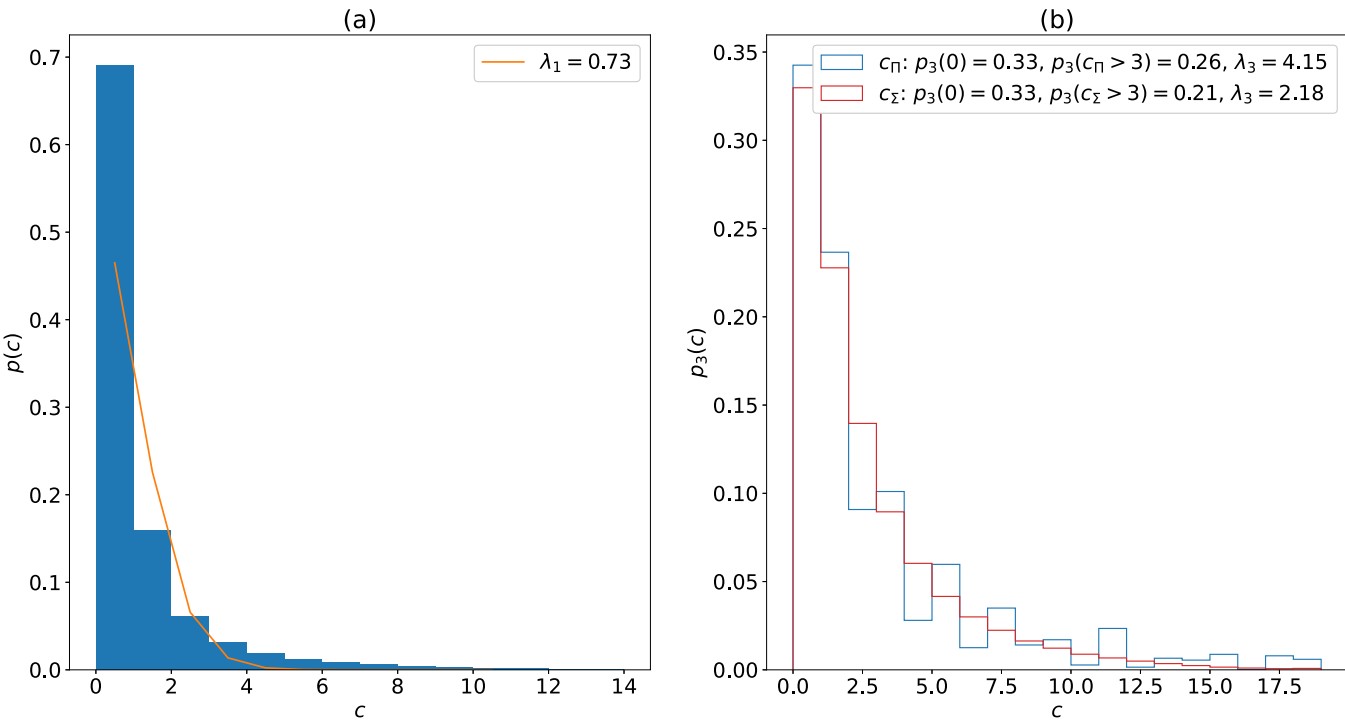

**Fig 1.** (a) Probability of a word having $c$ confusions. (b) Probability of a word triple having $c$ confusions.

address somewhere on the globe. Fig 1(b) also shows that

$$p_3(c_\Pi > 3) = 0.26, \qquad p_3(c_\Sigma > 3) = 0.21, \tag{5}$$

That is, around a quarter of addresses have more than three confusions, so at least one of the potentially correct addresses won't be shown by AutoSuggest. The word list used by W3W may have been tuned to avoid common homonyms (though in practice many words with homonyms can be found), but it does not seem to be tuned in any way to avoid typing errors, indeed there are many cases where a word and its plural appear. The number of addresses with more than 3 confusions is likely to be at least this high, especially if other error modes are considered or the frequency threshold lowered. An analysis of the dependency of $p_3(c_\Pi > 3)$ on the size of the 'common' set is given in Appendix A in S1 File which suggests that the above results are likely to be lower bounds/best cases for the number of addresses with more than three confusions globally.

## Confusing addresses locally

The primary address confusion issue discussed by W3W [25] and critics [9, 10] is the possibility of similar addresses occurring close together. For example, two potentially confusing addresses on opposite sides of a valley or lake could be disastrous for rescue workers. I will say that address $s$ **confusable** with address $s'$ if $s' \in C(s)$. It is shown in Appendix B in S1 File that, assuming a truly random distribution of addresses, the probability of at least one confusable pair in a circle of radius $r$ is

$$P(\hat{A}) \simeq 1 - \exp\left(-\frac{ca(r)^2}{2T}\right) \tag{6}$$

Where $T \simeq L^3$, $c$ is the average number of addresses confusable with any address and $a(r) = \pi (r/3)^2$ is the number $3m \times 3m$ grid cells within a radius $r$. Setting $c = 3$, $L = 40,000$ and $r = 4km$ gives $P(\acute{A}) \simeq 0.5$, i.e. to have a 50% probability to encounter a confusable pair requires a search radius of around 4km. This is quite a small error rate due to the very large factor $T \simeq L^3$ in the denominator. This number $T$, the size of the address space, is crucial for ensuring a small confusion probability. Because of the 'banding' one should perhaps use $T = K^3 \simeq (L/16)^3$ instead, giving $P(\acute{A}) \simeq 0.5$ at $r = 500m$. However, the 'effective' address space is in fact much smaller and the confusion probabilities much larger due to the details of the geocoding algorithm.

To understand this, first recall from that each address is equivalent to a number $m$, which is factored into a triple of integers e.g.

$$m = i^3 + (i+1)j + k$$

If values of $m$ are close, then the factorization will produce triples with the same first two numbers. Choosing an arbitrary run of consecutive $m$ values, these can be factored as follows

```
m                i       j       k
19999997500     (683,    2714,   1586)
19999997501     (683,    2714,   1587)
19999997502     (683,    2714,   1588)
19999997503     (683,    2714,   1589)
```

The $i, j, k$ values are used as indices in the word list, so the above could become something like

```
m                i        j        k
19999997500     (often,   home,    sea)
19999997501     (often,   home,    will)
19999997502     (often,   home,    see)
19999997503     (often,   home,    well)
```

With only one word different, the potential for confusion is much higher and the address space is of size $L$ rather than $L^3$. Similar conclusions apply for shared $j, k$ and $i, k$ as well as addresses sharing a single word in common.

To prevent this from happening W3W use a linear congruence to 'mix-up' the address indices, Eq 3. Linear congruence is a common way to introduce pseudo-randomness in applications, but is known to have many undesirable features which make the outputs non-random, e.g. [28]. For simplicity looking only at band 0, where $c_b = 0$

$$m = an \bmod b$$

$n$ is computed directly from the lat, lon values using Eq 2 and $a$, $b$ are band specific constants. I will take the numbers given in [14]: $a = 3,639,313$, $b = 20,000,000,000$. Consider a shift $\Delta m$ in the value of $m$ caused by a shift $\Delta n$ in the $n$ value.

$$m + \Delta m = a(n + \Delta n) \bmod b$$

this implies

$$\Delta m = a\Delta n \bmod b$$

So, when $a\Delta n$ is close to a multiple of $b$, $\Delta m$ will be small, meaning these two addresses are likely to have words in common. For the specific values of $a$ and $b$ above, if $\Delta n = 5,083,377$ then $a\Delta n \bmod b = \Delta m = 1$. Given there are around $1,485,706$ addresses per cell (at the latitude

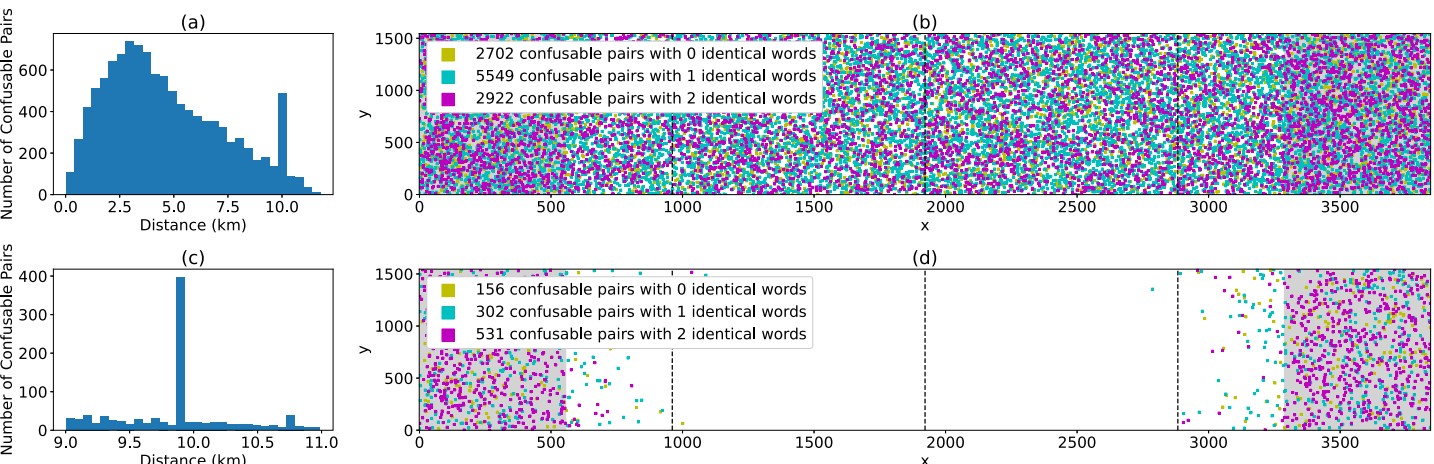

**Fig 2.** Top row: (a) Distribution of distances between confusable pairs across four adjacent cells. (b) Points are addresses with a confusion in one of the 4 cells (size exaggerated for clarity). Colour indicates how many words are shared between an address and its confusion. Dashed lines indicate *XY*-cell boundaries. Shaded areas indicate regions separated by $\Delta n$ = 5, 083, 377. Bottom row: As top but restricted to only show confusions between 9 and 11 km apart. (c) is the distribution of distances which peaks at 9.9km, corresponding to $\Delta m$ = 1. (d) Shows the disproportionate number of confusions with exactly two words in common between these zones.

of London, UK), this value of $\Delta n$ will be encountered about 3 cells away, which is not particularly far (cell size in London is around 4.5km by 3km) and highly likely to also be in band-0. This means, for any cell, a nearby cell will contain a long run of addresses which differ from an address in that cell by only a single word, greatly increasing the chances of confusion. $\Delta m = 1$ is the worst case but small $\Delta m$ values occur for any *a*, *b* pair and lead to scenarios where runs of addresses in nearby cells share many common words. These effects reduce the effective address space from $L^3$ to something significantly smaller and lead to many confusable pairs.

To see this I simulated four W3W cells with $Y = 3396$, $X = 4316$ and $q = 0$. In the first cell I found 685 confusable pairs. Simulating four neighbouring cells with $q = i \times 1, 485, 706$ for $i \in \{0, 1, 2, 3\}$ results in a roughly twenty fold increase in the number of confusable pairs to 11,173. Fig 2(a) shows the distribution of distances between confusable pairs across 4 band-0 cells, covering an area of roughly $14.5km \times 4.5km$. Many are quite close together. Fig 2(b) shows the locations of these confusable pairs (size exaggerated for visibility), coloured according to if there are $N_e = 0$, 1 or 2 words shared and in identical position between confusable addresses. As seen in (a) and highlighted in (c) and (d) there is a significant peak in confusable pairs separated by around 10km, which corresponds to $\Delta m = 1$.

As well as this peak, there are a large number of other confusable addresses within these 4 cells. $\Delta m = 1$ is the most extreme case, but any $\Delta n$ giving a small $\Delta m$ will reduce the effective address space. One should also consider $\Delta n$ values that connect confusable rather than identical words, for example runs of addresses like await.*v*.*w* and awaits.*v'*.*w'*. As many potential word confusions exist, there are many ways to make these problematic pairs, and this will be the case for any constants *a* and *b*. Thus the effective address space is much less than $L^3$ or even $K^3$ and Eq 6 grossly underestimates the number of confusing address pairs.

## Conclusions

This report has two main findings

1. Most of the simulated W3W addresses have one or more word triples with which they could be confused. The AutoSuggest feature goes some way towards remedying this, but

returns only 3 suggestions at a time and can only be restricted by country rather than say, region or county. I estimate 20–25% of addresses have more than 3 other addresses with which they could be confused, considering only homophones and typing errors. Thus it may be very difficult, if not impossible, to decipher a miscommunicated address with the current system.

2. The simulations performed suggest that the likelihood of nearby confusable pairs is far in excess of the very low (1 in 2.5 million [29]) likelihood quoted by W3W. This is due to the address assignment algorithm, in particular the use of linear congruence for randomising the addresses.

The solution to the first problem is straightforward—increase the number of search results and/or allow the user to restrict the search to specific geographic regions. The AutoSuggest algorithm could also be improved to account for some of the other possible error types e.g. word transposition. The second problem is trickier, and has been pointed out before in [9], where many nearby confusable pairs were found. Since the W3W algorithm is proprietary and their tools, including the word list, are closed source, there is little opportunity for researchers to fix any of these issues. W3W ambiguity could be resolved by transmitting multiple addresses or spelling out the words, but then the main advantage of W3W, simplicity, is lost. Also in some cases e.g. a lost, injured hiker, this might not be possible.

What3Words is an interesting attempt at an addressing system that doesn't use alphanumeric codes and is non-hierarchical. The use of words instead of alphanumeric codes is novel but, as shown, has high potential for ambiguity. It may be argued that alphanumeric codes have similar potential for confusion, however long developed practice reduces those issues e.g. UW63ABC → UW 63 ABC → "Uniform-Whiskey" "sixty-three" "Alpha-Bravo-Charlie". Alphanumeric codes could also be much easier for non-native speakers and are less likely to be affected by accent and/or difficulties in pronunciation, since the sender and receiver are only using tokens from a very restricted set.

W3W addresses being non-hierarchical is crucial for minimising address confusion, but comes with some drawbacks. For example, it is obvious that the PlusCodes [30] `PFQ9+6P7, PFQ9+6P8, PFQ9+6P9` are close together, while it is far from obvious that *washed.expand. logs, poems.joke.urban, punks.nails.villa* are neighbours. Being able to spot and use such patterns can be very useful for e.g. data analysts or, to go back to the lost hiker example, systematic searching. W3W has a terms of service which prohibits storing or sharing addresses and coordinates together under certain conditions ([18] section 4.3 and 5.5). In the worst case, if a large number of W3W addresses are recorded to be analysed later, this will require expensive and time consuming 'rehydration' using paid APIs. If only coordinates are saved, then this may require deleting the raw data, namely the W3W addresses themselves. A thorough exploration of these issues is beyond the scope of this work, but should be considered in any analysis of W3W for specific use cases.

There are numerous alternative addressing systems, to name a few of the more prominent ones: GeoHash [31], Natural Area Codes [32], Plus Codes (also known as Open Location Code) [30], the Military Grid Reference System [33] (for military applications) and the Maidenhead Locator System [34] (for amateur radio), which are potential alternatives to W3W. Some of these are quite well developed and supported, for example, PlusCodes are integrated into Google Maps and are made more user-friendly by substituting the top level information with standard place names and keeping the lower level for fine detail e.g. a PlusCode for the Exeter Computer Science Department is: UK, Exeter, `PFQ9+6P7`. For the lost hiker scenario I would suggest the OS-Grid Reference system as an alternative for UK applications. This is open-source, widely understood and, critically, printed on most maps. Yet another potential

issue is that representing W3W addresses on paper maps would be quite difficult! Such fail-safes are necessary in emergency applications, where issues like dead batteries or water damage to phones and devices must be considered.

Contrary to what W3W claim [35], this work has shown that many confusable W3W addresses are likely to exist and has identified some serious problems with the W3W algorithm. However, while showing that many confusable pairs exist, this work does not show that they *will* be confused. If W3W, or any addressing system, is to become critical infrastructure, relied upon by emergency services and other key agencies, then claims for ease of use and reliability should be thoroughly examined and empirically compared to alternatives, from standard street addresses to post codes to newer systems, like PlusCodes. There is a large academic literature dedicated to evaluating, comparing and improving geocoding systems e.g. [1, 36–40] which can provide guidance here. Until such an evaluation has been carried out I would caution against the widespread adoption of W3W, especially in applications with a noisy communication channel between sender and receiver. This work has tested some of the claims that W3W makes about their algorithm. These claims are not borne out under scrutiny which suggests that W3W should not be adopted as critical infrastructure without a thorough evaluation against a number of competing alternatives.

## Supporting information

**S1 File.**
(PDF)

## Author Contributions

**Conceptualization:** Rudy Arthur.

**Data curation:** Rudy Arthur.

**Formal analysis:** Rudy Arthur.

**Investigation:** Rudy Arthur.

**Methodology:** Rudy Arthur.

**Software:** Rudy Arthur.

**Visualization:** Rudy Arthur.

**Writing – original draft:** Rudy Arthur.

**Writing – review & editing:** Rudy Arthur.

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
