## [Decision Letter · Decision Letter 0]

15 Aug 2023

PONE-D-23-18725Studying the What3Words Geocoding SystemPLOS ONE

Dear Dr. Arthur,

Thank you for submitting your manuscript to PLOS ONE. After careful consideration, we feel that it has merit but does not fully meet PLOS ONE’s publication criteria as it currently stands. Therefore, we invite you to submit a revised version of the manuscript that addresses the points raised during the review process.

We look forward to receiving your revised manuscript.

Kind regards,

Hong Qin

Academic Editor

PLOS ONE

5. We note that you have referenced (Wood N. Autocorrect awareness: Categorizing autocorrect changes and measuring authorial perceptions) which has currently not yet been accepted for publication. Please remove this from your References and amend this to state in the body of your manuscript: (ie “Bewick et al. [Unpublished]”) as detailed online in our guide for authors

Reviewers' comments:

Reviewer's Responses to Questions

**Comments to the Author**

1. Is the manuscript technically sound, and do the data support the conclusions?

Reviewer #1: Yes

Reviewer #2: Yes

2. Has the statistical analysis been performed appropriately and rigorously? 

Reviewer #1: Yes

Reviewer #2: Yes

3. Have the authors made all data underlying the findings in their manuscript fully available?

Reviewer #1: Yes

Reviewer #2: Yes

4. Is the manuscript presented in an intelligible fashion and written in standard English?

Reviewer #1: Yes

Reviewer #2: Yes

5. Review Comments to the Author

Reviewer #1: Overall:

This is generally a well-written and well-organised paper that assesses the reliability of the What3Words geocoding application. The paper applies a convincing methodology to explore the potential for confusing different addresses – despite the W3W algorithm not being open-source. I have read the paper with great pleasure, and I only have some minor issues which I think should be addressed before publication.

Comments:

- Section 1 is a little confusing as it is explained that W3W uses a closed-source algorithm (p. 1) and when the aim of the paper is subsequently introduced it does not become immediately clear how it is possible to analyse the algorithm.

- The paper quite heavily relies on the idea that because W3W is widely adopted by emergency services, it needs to minimise any source of confusion (Section 1). This is supported by a reference that 85% of emergency services in the UK use W3W as well as that there is a website devoted to W3W issues. However, the “85%” is a claim made by W3W themselves (without a reference) and the dedicated website has not been updated in over a year – it also only received a handful of updates in 2022. The paper would benefit from some further justification or further statistics, where possible, on the general adoption of the W3W application (i.e. in Section 1).

- The description of the W3W application (Section 2) is very technical, which is fine, but it would benefit from a bit more of a general explanation how a user would interact with the application, for instance, in case of an emergency. I think a simple map showing part of the grid with associated words would be beneficial before moving on to the details of the algorithm.

- The description of how the word list is created (Section 3.3) could be improved in clarity. Why is ‘D(150)’ selected rather than, for instance, ‘D(1000)’ or ‘(D200)’. Perhaps some sensitivity analysis would be helpful as this seems to have quite an impact on the size of the confusion set?

Minor comments:

- Table 1: ‘tree’ and ‘three’ are homophones my accent but not in others. This sentence seems to be incomplete.

- “However in practice many can be found and, as mentioned in Table 1, homophony is dependent on accent and it is unlikely that all homophones across all regional accents, even in the UK, can be removed.” (p.4). Is there any way to include a (rough) estimate on how many homophones are likely to be included in the W3W word list?

- “…only lower case words which appear at least 3 times in the corpus of [19] are kept.” (p.5). It is not clear to me how words can appear multiple times in a frequency-based word list?

- “Generate the set of all strings that could be formed by making the typing errors listed above to make Ct(w).” (p.5). How exactly?

- Figure 2b is not really clear and I find it difficult to interpret. Perhaps this would work better for a smaller area?

Reviewer #2: PONE-D-23-18725

Studying the What3Words Geocoding System

Review

The paper presents an initiative to assess important characteristics of the What3Words (W3W) geocoding system, in face of anecdotical evidence of confusion involving 3-word addresses, composed of common English language words, provided by the system.

Since W3W is not open software, nor does it open its reference data (specially the list of words used to compose the 3-word addresses), a direct investigation would require executing a large number of accesses, in order to gather actual W3W data. These accesses have a cost, so the author instead devised a strategy to simulate the W3W algorithm and list of words, in order to perform the intended analyses. Furthermore, as the terms of use do not allow storing addresses along with their coordinates, even if large numbers of accesses were free, an external analyst would have trouble in devising a fair assessment method.

As a result, readers should keep in mind that the analyses presented in the paper do not actually refer to W3W as an online service, but to a simulation of W3W based on what is known about its basic working logic, algorithms, and word selection strategy. The title and many statements along the text should reflect this fact. The paper’s title, for instance, could be changed to something like “A simulation and critical analysis of the What3Words geocoding system”.

In the path towards creating a W3W simulation, the author makes a number of assumptions, especially on the constitution of the list of words that are indexed by the method. In the paper, a text corpus is used to obtain a list of English words along with their frequencies, from which words with less than 4 characters are discarded and the most common remaining ones are selected. The actual selection process in W3W is unknown, but (again) anecdotical evidence is used to show a number of issues, such as the use of plurals, similar-sounding groups of words, spelling variations and more.

The paper shows the occurrence of the types of confusion that might happen in actual W3W addresses, and demonstrates that they occur quite frequently in the simulated setting, with the selection of words devised for this study (not the actual W3W list of words). Given the size of the corpus, the number of words in the list, and the publicly known characteristics of W3W, the simulation seems fair, but its conclusions are more directly applicable to the simulation, rather than to the actual service. Conclusions, therefore, are indirect – and, of course, there would be no other way to draw more precise conclusions, other than by having access to the actual W3W code and data, or maybe a large number of queries to the system.

The indirect conclusions are not a problem per se, but naturally there remain some additional doubts regarding their applicability to the actual service. The qualitative assessments presented as conclusions could be strengthened if verified using the actual service. This, of course, would require acquiring a license to use W3W for some time period. I list below a few possible strategies for doing so:

1. Randomly select W3W addresses, and verify their vicinity in search for nearby confusing addresses up to some maximum distance. Verify if the results fall within the probability determined in the simulation.

2. Generate a number of confusing addresses from the simulation data, and check for the existence of the confusing versions in actual W3W; then verify their distance.

Example: small.dark.places (mentioned as an example in the paper) exists in Edinburgh, UK; dark.small.places exists in Illinois, USA. Large distance, not a problem.

3. Pick a pair of words for the first two positions, and then vary the third using words from the simulation list. Analyze the spatial distribution of the valid results.

Given, however, the stated limitations of this study, the simulation approach is adequate and the paper effectively shows weak points and potential problems with the W3W algorithm and related data. I suggest revising the phrasing on some claims, assessments and estimations, in order to make it clear the extent to which the simulation is expected to accurately replace the behavior of the actual system.

I also would like to point out the importance of making these evaluations widely known since, while agreeing with the author’s admonitions against the use of W3W by voice over noisy channels, I can clearly see that W3W can be improved based on the systematic characterization of its potential problems. Various alternatives come to mind, but I’ll refrain from mentioning them here, as solutions are not within the purview neither of the paper, nor of the review.

Localized issues, questions, suggestions:

1. Abstract: … for being less reliable THAN claimed (typo)

2. Section 4: W3W’s claim on the overwhelming proportion of similar-sounding word combinations being far apart cannot actually be verified in the simulation, so I suggest rephrasing the second sentence. Also, your observation that the autocorrection feature “seems to balance address similarity with prior information about places” should be better explained. Did you verify that suggestions are of addresses near the user?

3. The lost hiker scenario could possibly be resolved if the hiker transmits two W3W addresses, e.g. for neighboring cells. Chances of a mistake in both of them at the same time, directing responders to a different nearby place with the two distortions close to each other should be very low. Of course, transmitting too much defeats the purpose of using W3W in the first place: it then could be better to spell out each digit of the lat and lon coordinates, or to spell out each letter of each of the three words using ICAO (alpha-bravo--charlie-)…

4. Page 16, first paragraph: “I generated three W3W cells…” -> “I simulated the composition of three W3W cells using the word list from Section 3.”

5. Conclusions: “Most W3W addresses have one or more word triples with which they could be confused” -> Most of the simulated W3W addresses tend to have one or more triples… or Simulation results indicate that addresses created using the W3W method have one or more…

6. Conclusions: “is far in excess of” -> simulation indicates that the likelihood of…

7. Page 17, “Departement”-> Department (? typo)

8. Conclusions: “…this work has shown that many confusable W3W addresses exist…” -> many confusable W3W addresses are likely to exist.

9. Conclusions: “This work has tested some of the claims that W3W makes about their algorithm. These claims are not borne out under scrutiny which suggests that W3W should not be adopted as critical infrastructure without a thorough evaluation against a number of competing alternatives.” I totally agree with these statements.

6. PLOS authors have the option to publish the peer review history of their article (what does this mean?). If published, this will include your full peer review and any attached files.

Reviewer #1: No

Reviewer #2: **Yes: **CLODOVEU A DAVIS JR

---

## [Author Response · Author response to Decision Letter 0]

19 Aug 2023

I have included my responses to the queries raised in the "Response to Reviewers" document and updated the MS accordingly.

---

## [Decision Letter · Decision Letter 1]

22 Sep 2023

A Critical Analysis of the What3Words Geocoding Algorithm

PONE-D-23-18725R1

Dear Dr. Arthur,

We’re pleased to inform you that your manuscript has been judged scientifically suitable for publication and will be formally accepted for publication once it meets all outstanding technical requirements.

Kind regards,

Hong Qin

Academic Editor

PLOS ONE

Additional Editor Comments (optional):

Reviewers' comments:

Reviewer's Responses to Questions

**Comments to the Author**

1. If the authors have adequately addressed your comments raised in a previous round of review and you feel that this manuscript is now acceptable for publication, you may indicate that here to bypass the “Comments to the Author” section, enter your conflict of interest statement in the “Confidential to Editor” section, and submit your "Accept" recommendation.

Reviewer #1: All comments have been addressed

Reviewer #2: All comments have been addressed

2. Is the manuscript technically sound, and do the data support the conclusions?

Reviewer #1: Yes

Reviewer #2: Yes

3. Has the statistical analysis been performed appropriately and rigorously? 

Reviewer #1: Yes

Reviewer #2: Yes

4. Have the authors made all data underlying the findings in their manuscript fully available?

Reviewer #1: Yes

Reviewer #2: Yes

5. Is the manuscript presented in an intelligible fashion and written in standard English?

Reviewer #1: Yes

Reviewer #2: Yes

6. Review Comments to the Author

Reviewer #1: The author has been very responsive and I am happy with the changes that have been made to the manuscript. The worked example on how the word list was created is very helpful and the changes to Figure 2b are great and (again) very helpful. I also appreciate the efforts in conducting a sensitivity analysis to support the D(150) threshold. I am looking forward to seeing the article in print.

Reviewer #2: The paper presents an initiative to assess important characteristics of the What3Words (W3W) geocoding system, in face of anecdotical evidence of confusion involving 3-word addresses, composed of common English language words, provided by the system. The analysis cleverly avoids the closed parts of the W3W system, and simulates them according to information available on the patent. The result is convincing, and presents a sound discussion on specific aspects of the algorithm and service. The conclusions may be used as a set of guidelines to improve on W3W ideas, possibly to develop a better alternative system in the future.

7. PLOS authors have the option to publish the peer review history of their article (what does this mean?). If published, this will include your full peer review and any attached files.

Reviewer #1: No

Reviewer #2: **Yes: **Clodoveu Davis

---

## [Editor Report · Acceptance letter]

26 Sep 2023

PONE-D-23-18725R1 

A Critical Analysis of the What3Words Geocoding Algorithm 

Dear Dr. Arthur:

I'm pleased to inform you that your manuscript has been deemed suitable for publication in PLOS ONE. Congratulations! Your manuscript is now with our production department. 

Kind regards, 

on behalf of

Dr. Hong Qin 

Academic Editor

PLOS ONE